# SuperCAT: Super Resolution and Cross Semantic Attribute-guided Transformer based Feature Refinement for Zero-Shot Remote Sensing Scene Classification

## Abstract

Zero-shot learning becomes challenging in classifying scenes of unseen classes due to the typical characteristics of remote-sensing images. The intricate variations and non-uniform spatial resolutions among the scenes of remote sensing images further complicate achieving discriminative semantic knowledge. To tackle these issues, we propose a SuperCAT framework comprising a super-resolution module, a cross-semantic attribute-guided Transformer (CAT), feature-generating models, and a feature refinement (FR) module for the zero-shot scene classification in remote sensing images. First, we leverage the semantic attributes for all the classes of three benchmark remote sensing scene classification datasets to explore semantic knowledge using super-resolution effectively. Then, the semantic attribute → visual Transformer (SAVT) and visual → semantic attribute Transformer (VSAT) modules in CAT learn to obtain attribute-based visual features and visual-based attribute features, respectively. The SAVT and VSAT modules collaboratively learn and teach each other using the feature-level and prediction-level semantic collaborative losses. The feature-generating models map semantic vectors to the visual features of remote-sensing images. The FR module incorporates triplet center margin loss and semantic loop consistency loss functions to capture class-related and semantically-related discriminative features for achieving intra-class closeness and inter-class distinctiveness. Our extensive experiments on three benchmark remote sensing image scene classification datasets demonstrate the efficacy of SuperCAT over state-of-the-art approaches. The code can be accessed at https://github.com/ZSL-RSI-SC/SuperCAT

## 1 Introduction

The field of remote sensing technology has witnessed remarkable advancements in collecting vast volumes of high-resolution earth observation data (Chi et al., 2016). Remote sensing images, in general, exhibit diverse objects with varying spatial configurations and non-uniform backgrounds. Scene classification helps understand large-scale remote-sensing images by partitioning them into multiple small patches or scenes. Each scene is labelled from predefined classes by analysing its content. The works on scene classification (Cheng et al., 2017b; 2020a) have shown progress by leveraging convolutional neural networks (CNNs). As remote sensing samples of new classes gradually emerge, these methods will not be able to recognize them unless the samples of new classes are considered during training. Also, collecting annotated scenes of remote sensing images for all the new classes is tedious and time-consuming. This motivates us to explore zero-shot learning for scene classification in remote-sensing images.

Zero-shot learning (ZSL) (Larochelle et al., 2008) is inspired by human recognition capabilities, aiming to recognize new classes by utilizing the shared semantic information from seen to unseen categories. In ZSL, samples from only seen classes are available during the learning phase, with no access to unseen classes. More precisely, the training and testing samples are distinct. The most common settings of zero-shot learning are conventional (CZSL) and generalized (GZSL) zero-shot

learning. The CZSL learns to classify only unseen categories, whereas GZSL classifies unseen and seen categories (Xian et al., 2017a).

In general, the scenes of remote sensing images exhibit unique characteristics in comparison to natural images. Further, the subtle differences among the scenes of unseen classes in ZSL add complexity to achieving discriminative semantic knowledge. Existing approaches are ineffective in addressing the cross-dataset bias because they rely on pre-trained models from ImageNet. Generally, the images in the ImageNet dataset depict the objects captured by a photographer from the side/front view. In contrast, remote sensing images represent the top view of objects on the ground, usually acquired through remote sensing platforms flown at high altitudes. Thus, the analysis of remote-sensing images needs different strategies compared to natural images due to many issues Cheng et al. (2020b), such as i) immense intraclass diversity, ii) high interclass similarity, iii) significant variance of scene/object scales, and iv) coexistence of multiple ground objects.

We leverage semantic attributes Rambabu et al. (2024) across three remote sensing benchmark datasets to capture the distinct characteristics of diverse scenes in zero-shot scene classification. By leveraging these semantic attributes, we propose a SuperCAT framework to effectively classify unseen and seen classes for zero-shot remote-sensing scene classification (ZSRSSC) tasks. Our proposed SuperCAT framework innovatively combines a super-resolution technology with the ZSRSSC task. The core of SuperCAT is a cross-semantic attribute-guided Transformer (CAT) module, which extracts visual features guided by semantic attributes, and simultaneously extracts semantic features guided by visual features. The SuperCAT facilitates learning by mapping semantic-to-visual correspondences and synthesizing features to build an efficient classifier. We leverage f-VAEGAN to map semantic vectors to visual representations. Further, SuperCAT employs a feature refinement (FR) module to enhance the visual features of both seen and unseen class samples in remote sensing images, optimizing classification performance in zero-shot learning scenarios.

In summary, our essential contributions are:

- We propose a SuperCAT framework that innovatively combines *super-resolution* with the zero-shot scene classification task to improve the classification performance of remote sensing images.

- We leverage the *semantic attributes* for three remote-sensing scene classification benchmarking datasets to explore the semantic knowledge in zero-shot scene classification.

- A *cross-semantic attribute-guided Transformer (CAT)* module is proposed to obtain attribute-based visual features and visual-based attribute features.

- We explore the feature generating (*f-VAEGAN*) and feature refinement (*FR*) modules to refine the visual features for zero-shot scene classification in remote sensing images.

- Extensive experiments and comparisons with state-of-the-art methods demonstrate the efficacy of the proposed SuperCAT framework in zero-shot remote scene classification tasks.

The rest of this paper is organized as follows. Section 2 introduces our SuperCAT framework. Section 3 presents the experimental results and an analysis of SuperCAT. Section 4 concludes this paper.

## 2 PROPOSED SUPERCAT FRAMEWORK

The block diagram of the proposed SuperCAT framework for zero-shot scene classification in remote sensing images is shown in Figure 1(a). The SuperCAT comprises a super-resolution module, a cross-semantic attribute-guided Transformer (CAT) Chen et al. (2021a) module, feature generating models (f-VAEGAN) (Xian et al., 2019), a feature refinement (FR) module (Chen et al., 2021b), and a classifier (CLS). Initially, we use ResShift (Yue et al., 2023), an efficient diffusion model, to obtain super-resolution images of remote sensing samples. Then, we extract visual features for each input super-resolution image through a ResNet101 CNN Backbone pre-trained on ImageNet. A word vector is generated for the corresponding semantic attributes using the word2vec (Mikolov et al., 2013) method. The CAT in the proposed SuperCAT comprises semantic attribute $\rightarrow$ visual Transformer (SAVT) and visual $\rightarrow$ semantic attribute Transformer (VSAT) to extract visual features guided by semantic attributes and semantic attribute features guided by visual features, respectively.

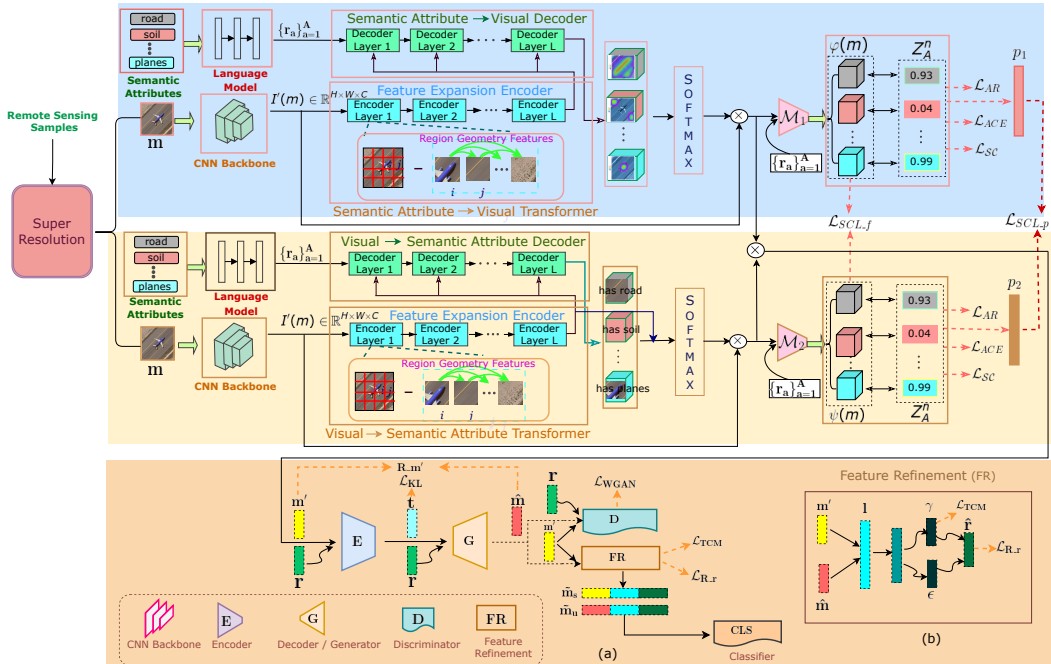

Figure 1: (a) The proposed SuperCAT framework block diagram for zero-shot scene classification in remote sensing images. (b) The architecture of the feature refinement (FR) module.

We also employ a semantical collaborative learning technique to help SAVT and VSAT learn collaboratively and teach others. During the training phase, the f-VAEGAN learns to generate visual features from the class semantic vector $\mathbf{r}$ (e.g., $\mathbf{r} = 33 \times 21$ matrix for the UCM21 Yang & Newsam (2010) dataset). Further, we employ the feature refinement(FR) module combined with f-VAEGAN to obtain discriminative visual features. Specifically, the FR module is optimized using triplet center margin (TCM) loss and semantic loop consistency (SLC) loss (Chen et al., 2021b). Figure 1(b) describes the architecture of the FR module to enhance visual features for unseen and seen class examples. Finally, a classifier is learned to classify enhanced unseen and seen class features.

*Notation:* Let $N^u$ and $N^s$ be the sets of unseen and seen class samples, respectively. Seen class samples are denoted as $S_c = \{m_i^s, n_i^s\}$, where $m_i^s$ represents a visual feature, and $n_i^s$ is the respective class label $\in N^s$. Similarly, the unseen class samples are defined as $U_c = \{m_i^u, n_i^u\}$, where $m_i^u$ represents a visual feature, and $n_i^u$ is the respective class label belonging to $N^u$. For each $n \in N$, we have a set of semantic vectors comprising $A$ attributes denoted as $z^n = [z_1^n, ..., z_A^n]^T$. These semantic vectors help in transferring semantic information from seen to unseen classes. We obtain attribute vectors for each attribute $\mathcal{R}_A = \{r_a\}_{a=1}^A$ using the word2vec model Mikolov et al. (2013), applied to the words of attribute names (e.g., $\{r_a\}_{a=1}^A = 33 \times 300$ for the UCM21 dataset). In the context of ZSL, the task is to determine the class label $n^u \in N^u$ in the case of CZSL. In the GZSL setting, the goal is to identify class label $n \in N^s \cup N^u = N$, with the constraint that $N^s \cap N^u = \phi$.

## 2.1 SUPER-RESOLUTION

We use ResShift (Yue et al., 2023), an efficient diffusion model for super-resolution, to minimize the number of sampling steps. The ResShift model leverages a Markov chain to transition between high-resolution images and their corresponding low-resolution versions by constructing a transition kernel that gradually shifts the residual between them. This approach incorporates a flexible noise schedule designed to control both the shifting speed of the residual and the noise intensity at each step.

## 2.2 Cross semantic Attribute-guided Transformer (CAT) Module

This module (Chen et al., 2021a) comprises semantic attribute → visual Transformer (SAVT) and visual → semantic attribute Transformer (VSAT) submodules.

### 2.2.1 Semantic Attribute → Visual Transformer (SAVT):

The SAVT comprises a feature expansion encoder and a semantic attribute → visual decoder.

*Feature Expansion Encoder (FEE):* The FEE enhances the image features by mitigating cross-dataset bias between ImagNet & ZSRSSC benchmark datasets Chen et al. (2021b). Generally, feature vectors ($I'(m) \in \mathbb{R}^{H \times W \times C}$) obtained from CNNs inherently entangle the feature representations among different image parts, obstructing the transferability of semantic knowledge from seen to novel classes Xu et al. (2020). Hence, feature-augmented and scaled dot-product-based attention is proposed to improve the encoder by minimizing corresponding geometry associations from visual features. To obtain related geometry features Herdade et al. (2019); Zhang et al. (2021), we initially determine the related center positions ($p_i^{cen}, q_i^{cen}$) depending on the pair of 2D corresponding coordinates of the $i^{th}$ grid $\{(p_i^{min}, q_i^{min}), (p_i^{max}, q_i^{max})\}$:

$$(p_i^{cen}, q_i^{cen}) = \left( \frac{p_i^{min} + p_i^{max}}{2}, \frac{q_i^{min} + q_i^{max}}{2} \right), \tag{1}$$

$$w_i = (p_i^{max} - p_i^{min}) + 1, \tag{2}$$

$$h_i = (q_i^{max} - q_i^{min}) + 1, \tag{3}$$

where $(p_i^{min}, q_i^{min})$ and $(p_i^{max}, q_i^{max})$ are the corresponding coordinates of the top left corner & bottom right corner of the grid $i$, respectively. Later, a region geometry features $X_{ij}$ between grid $i$ & grid $j$ are created using:

$$X_{ij} = ReLU(w_r^T y_{ij}), \tag{4}$$

$$\text{where} \quad y_{ij} = FC(g_{ij}), \qquad g_{ij} = \begin{pmatrix} log\left( \frac{|p_i^{cen} - p_j^{cen}|}{w_i} \right) \\ log\left( \frac{|q_i^{cen} - q_j^{cen}|}{h_i} \right) \end{pmatrix}, \tag{5}$$

where $g_{ij}$ is the related geometry relation between grid $i$ & grid $j$, FC represents a fully connected layer, ReLU is used after the FC layer, and $w_r^T$ represents learnable weights.

Eventually, we neglect the region geometry features from the visual features of the feature-expanded scaled dot-product attention to give a better precise attention map, formulated as:

$$Q^e = I(m)W_q^e, K^e = I(m)W_k^e, V^e = I(m)W_v^e, \tag{6}$$

$$Z_{aug} = softmax\left( \frac{Q^e K^{e^T}}{\sqrt{d^e}} - X \right), \tag{7}$$

$$I_{aug}(m) \leftarrow I(m) + Z_{aug}, \tag{8}$$

where $V$, $K$, and $Q$ indicate value, key, and query matrices, respectively, $W_v^e$, $W_k^e$, $W_q^e$ denote learnable weight matrices, $d^e$ specifies the factor of the scaling, and $Z_{aug}$ indicates the augmented features. $I(m) \in R^{H \times W \times C}$ are the arranged image features obtained from the feature vectors embedded by an FC layer that succeeded by a ReLU and Dropout layer. $I_{aug}(m)$ represents the augmented visual features obtained from FEE. They will facilitate the following sequential learning. We rephrase the $I_{aug}(m)$ as $I_{aug}^{a \to v}(m)$ and $I_{aug}^{v \to a}(m)$ in SAVT and VSAT, respectively.

*Semantic Attribute → Visual Decoder (SAVD):* We employ a SAVD to obtain visual features based on semantic attributes by using the cross-attention operator Chen et al. (2021a), which focuses on visual features from attribute features. The decoding procedure continually includes visual features under the guidance of semantic attribute information $\mathcal{R}_A$. Hence, the SAVD can effectively position the image region with the utmost applicability for every attribute in a specified image. The encoder

layer outputs $I_{aug}^{a \to v}(m)$ are used as inputs to multi-head cross attention, as keys ($K_t^{a \to v}$), values ($V_t^{a \to v}$), and queries ($Q_t^{a \to v}$) to be obtained as semantic embeddings $\mathcal{R}_A$, formulated as:

$$Q_t^{a \to v} = \mathcal{R}_A W_{qt}^{a \to v}, \tag{9}$$

$$K_t^{a \to v} = I_{aug}^{a \to v}(m) W_{kt}^{a \to v}, \tag{10}$$

$$V_t^{a \to v} = I_{aug}^{a \to v}(m) W_{vt}^{a \to v}, \tag{11}$$

$$H_t = softmax\Big(\frac{Q_t^d K_t^{a \to v T}}{\sqrt{d^d}}\Big) V_t^{a \to v}, \tag{12}$$

$$\tilde{F} = ||_{t=1}^{T}(H_t) W_o^{a \to v}, \tag{13}$$

where $W_{qt}^{a \to v}$, $W_{kt}^{a \to v}$, $W_{vt}^{a \to v}$, and $W_o^{a \to v}$ specify weight matrices, $\sqrt{d^d}$ indicates a factor of scaling, $\tilde{F}$ represents the attribute-based visual features, and $||$ denotes a function of concatenation. Then, a ReLU after every two linear transformations of the feed-forward network (FFN) is applied over $\tilde{F}$, as:

$$F' = ReLU(\tilde{F} W_1^{a \to v} + b_1^{a \to v}) W_2^{a \to v} + b_2^{a \to v}, \tag{14}$$

where $b_1^{a \to v}, b_2^{a \to v}, W_1^{a \to v}, W_2^{a \to v}$ specify biases and weights of the layers in FFN correspondingly. $F' = \{F_1', \ldots, F_A'\}$ represents final visual features are based on attributes. Then, a softmax activation function is applied to $F'$, and the resultant feature dimension of $F'$ does not match the original visual feature dimension $M$ (e.g., 2048-dim feature vector extracted from ResNet101). Thus, $F'$ is transformed to attribute-based visual features $\tilde{m}$(with the same dimension of input feature) through the original visual features $M$ (to give input to the next stage) as:

$$F = \text{Softmax}(F'), \tag{15}$$

$$\tilde{m} = F \times M. \tag{16}$$

*Visual-Semantic Projection Network (VSPN):* The VSPN determines visual-semantic interactions by mapping the obtained attribute-based visual features to the semantic embedding space depending on the mapping function $\mathcal{M}_1$, which is given by:

$$\varphi(m_i') = \mathcal{M}_1(F) = \mathcal{R}_A^T W_3^{a \to v} F, \tag{17}$$

where $W_3^{a \to v}$ represents a projection matrix which projects $F$ to the semantic embedding space. The $\varphi(m_i')[r]$ represents the attribute score that specifies the confidence of attribute $r$ in the image $m_i$.

## 2.3 VISUAL → SEMANTIC ATTRIBUTE TRANSFORMER (VSAT)

Like SAVT, we employ a visual → semantic attribute Transformer to obtain visual-based attribute features that focus on the semantic attributes corresponding to every image region. Attribute-based visual features & visual-based attribute features are complimentary and calibrate each other to learn more intrinsic semantic information between them. Initially, like SAVT, VSAT uses the feature expansion encoder to enhance visual features as $I_{aug}^{v \to a}(f)$. Subsequently, these features are used in the visual → semantic attribute decoder of VSAT.

*Visual → Semantic Attribute Decoder (VSAD):* After visual feature enhancement, we employ a VSAD to learn visual-based attribute features. The operator of our cross-attention focuses on attributes from visual features Chen et al. (2021a), formulated as:

$$Q_t^{v \to a} = I_{aug}^{v \to a}(m) W_{qt}^{v \to a}, \tag{18}$$

$$K_t^{v \to a} = \mathcal{R}_A W_{kt}^{v \to a}, \tag{19}$$

$$V_t^{v \to a} = \mathcal{R}_A W_{vt}^{v \to a}, \tag{20}$$

$$H_t = \text{softmax}\Big(\frac{Q_t^d K_t^{v \to a T}}{\sqrt{d^d}}\Big) V_t^{v \to a}, \tag{21}$$

$$\tilde{U} = ||_{t=1}^{T}(H_t) W_o^{v \to a}, \tag{22}$$

where $W_{qt}^{v \to a}$, $W_{kt}^{v \to a}$, $W_{vt}^{v \to a}$, and $W_o^{v \to a}$ specify weight matrices, and $\tilde{U} = \{\tilde{U}_1, \ldots, \tilde{U}_K\}$ represents the set of visual-based attribute features. Inherently, the visual semantic representations $\tilde{U}$ are obtained corresponding to visual regions ($K = H \times W$) of an image. Then, a ReLU after every two linear transformations of FFN is performed over $\tilde{U}$, as given below:

$$U' = ReLU(\tilde{U}W_1^{v \to a} + b_1^{v \to a})W_2^{v \to a} + b_2^{v \to a}, \tag{23}$$

where $b_1^{v \to a}, b_2^{v \to a}, W_1^{v \to a}, W_2^{v \to a}$ specify biases and weights of the linear layers correspondingly, and $U'$ denotes the final visual-based attribute features. Then, a softmax function is applied to $U'$. The resultant feature dimension of $U$ does not match the original visual feature dimension ($M$). Thus, $U$ is transformed to visual-based attribute features $\bar{m}$ through the original visual features $M$ as:

$$U = \text{Softmax}(U'), \tag{24}$$
$$\bar{m} = U \times M. \tag{25}$$

*Visual-Semantic Projection Network (VSPN):* We map the visual-based attribute features to the semantic embedding space using the mapping function $\mathcal{M}_2$. We consider the augmented visual features $I_{aug}^{v \to a}(m)$ obtained from the FEE to encourage effective mapping. Initially $\mathcal{M}_2$ maps $U$ into $K$ region scores $\bar{U}$, formulated as:

$$\bar{U} = \mathcal{M}_2(U) = I_{aug}^{v \to a}(m)^T W_3^{v \to a} U, \tag{26}$$

where $W_3^{v \to a}$ represents a learnable projection matrix. Here, the dimension of $\bar{U}$ is K-D, which does not equal the dimension of class semantic vector A-D. Hence, we further map $\bar{U}$ into the semantic attribute space with the dimension of $A$ based on an attention score $Attn = \mathcal{R}_A^T W_{attn} I(m) \in \mathbb{R}^{A \times K}$ using the mapping function $\mathcal{M}_2$, where $W_{attn}$ represents a learnable embedding matrix given by:

$$\psi(m_i) = Attn \times \bar{U}. \tag{27}$$

Like $\varphi(m_i)$, the $\psi_r(m_i)$ represents the attribute score that specifies the confidence of a-th attribute confining to the image $m_i$. The optimization of the CAT module ($\mathcal{L}_{CAT}$) is discussed in the supplementary material.

## 2.4 CAT OPTIMIZATION

The SAVT and VSAT components employ the following three loss functions based on remote sensing attributes to optimise the CAT module.

*Attribute Regression Loss:* In the zero-shot remote sensing scene classification task, we consider visual-semantic interaction as a regression problem and reduce the mean square error between the original attribute score $z^n$ and predicted attribute score $x(m_i)$ for a batch of $n_b$ training samples $m_i^s$:

$$\mathcal{L}_{AR} = \frac{1}{n_b} \sum_{i=1}^{n_b} \|x(m_i^s) - z^n\|_2^2, \tag{28}$$

where $x(m_i^s) = \varphi(m_i^s)$ for SAVT and $x(m_i^s) = \psi(m_i^s)$ for VSAT.

*Attribute-based Cross-entropy Loss:* When a remote sensing attribute is visually available in an image, the corresponding visual feature is perfectly projected near the semantic class vector $z^n$. $\mathcal{L}_{ACE}$ is defined given below from the given $n_b$ training samples $\{m_i^s\}_{i=1}^n$ with their respective semantic class vectors $z^n$:

$$\mathcal{L}_{ACE} = -\frac{1}{n_b} \sum_{i=1}^{n_b} log \frac{exp(x(m_i^s) \times z^n)}{\sum_{\hat{n} \in N} exp(x(m_i^s) \times z^{\hat{n}})} \tag{29}$$

*Self-calibration loss:* The CAT module is certainly biased to seen classes since $\mathcal{L}_{AR}$ and $\mathcal{L}_{ACE}$ are optimized only on seen classes Zhu et al. (2019); Xu et al. (2020). We employ a self-calibration loss

($\mathcal{L}_{SC}$) to overcome this issue to move certain predicted probabilities from the seen classes to the unseen. $\mathcal{L}_{SC}$ is defined as:

$$\mathcal{L}_{SC} = -\frac{1}{n_b} \sum_{i=1}^{n_b} \sum_{n'=1}^{N^u} log \frac{exp(x(m_i^s) \times z^{n'} + \mathbb{I}_{n' \in N^u})}{\sum_{\hat{n} \in N} exp(x(m_i^s) \times z^{\hat{n}} + \mathbb{I}_{\hat{n} \in N^u})} \tag{30}$$

where $\mathbb{I}_{n \in N^u}$ indicates an indicator function (i.e. $\mathbb{I} = 1$ when $n \in N^u$, else 0 ).

*Semantical Collaborative Learning:* Further, we employ feature-level ($\mathcal{L}_{SCL\_f}$) and prediction-level ($\mathcal{L}_{SCL\_p}$) semantic collaborative loss functions to assist SAVT & VSAT to collaboratively learn from each other throughout the learning stage for CAT optimization. We have used an $l_2$ distance to implement these two losses. Especially, we have utilised an $l_2$ distance between the semantically enriched visual features of SAVT and VSAT for a given test scene image $m_i$, formally defined as:

$$\mathcal{L}_{SCL\_f} = \frac{1}{n_b} \sum_{i=1}^{n_b} \left\| \varphi(m_i^s) - \psi(m_i^s) \right\|_2^2. \tag{31}$$

Similarly, we also used an $l_2$ distance between the predictions of the SAVT and VSAT (i.e., $p_1$ and $p_2$), defined as:

$$\mathcal{L}_{SCL\_p} = \frac{1}{n_b} \sum_{i=1}^{n_b} \left\| p_1(m_i^s) - p_2(m_i^s) \right\|_2^2. \tag{32}$$

The components SAVT and VSAT are trained with three loss functions, i.e., $\mathcal{L}_{SC}, \mathcal{L}_{ACE}$, and $\mathcal{L}_{AR}$, formally defined as:

$$\mathcal{L}_{SAVT} = \lambda_{AR}\mathcal{L}_{AR}^{SAVT} + \mathcal{L}_{ACE}^{SAVT} + \lambda_{SC}\mathcal{L}_{SC}^{SAVT}, \tag{33}$$

$$\mathcal{L}_{VSAT} = \lambda_{AR}\mathcal{L}_{AR}^{VSAT} + \mathcal{L}_{ACE}^{VSAT} + \lambda_{SC}\mathcal{L}_{SC}^{VSAT}, \tag{34}$$

where the hyperparameters $\lambda_{AR}$ and $\lambda_{SC}$ help control their loss functions in the SAVT and VSAT. Lastly, we define the total loss function for the CAT module:

$$\begin{aligned} \mathcal{L}_{CAT} = &\lambda_{SAVT}\mathcal{L}_{SAVT} + \lambda_{VSAT}\mathcal{L}_{VSAT} \\ &+ \lambda_{SCL\_f}\mathcal{L}_{SCL\_f} + \lambda_{SCL\_p}\mathcal{L}_{SCL\_p}, \end{aligned} \tag{35}$$

where $\lambda_{SCL\_f}$, $\lambda_{SCL\_p}$ and $\lambda_{VSAT}$ represent the parameters to control their respective loss functions. We set the $\lambda_{SAVT}$ to one to stabilise the CAT during the training stage.

The attribute-based visual features $\tilde{m}$ and visual-based attribute features $\bar{m}$ obtained from the CAT module are separable under softmax loss supervision. However, they lack the discriminative power capability for accurately predicting the labels of unseen classes in remote sensing scene classification, showing immense intraclass diversity and high interclass similarity. Consequently, utilizing these features directly to recognise unseen classes may not be ideal. Hence, we combine these feature vectors $m' = \tilde{m} \odot \bar{m}$ and refine them to enhance the feature separability and efficient label prediction of unseen classes.

## 2.5 Feature Generating Models

Most generative-based zero-shot learning methods use f-VAEGAN to generate synthetic CNN features while adhering to the semantic vector $r$ constraints in transforming semantic attribute vectors into visual features. We also employ the f-VAEGAN (Xian et al., 2019), comprising a feature-generating VAE (f-VAE) and a feature-generating network (f-WGAN). The f-VAE has two key components: an encoder $E(m', r)$ and a conditional generator $G(t, r)$ from f-WGAN, which acts as a decoder $G$. The encoder $E$ transforms an input $m'$ into hidden features $t$, and the decoder $G(t, r)$ reconstructs the input feature $\hat{m}$ from $t$. The optimization of f-VAE can be expressed as follows:

$$\begin{aligned} \mathcal{L}_{VAE} &= \mathcal{L}_{KL} + \mathcal{L}_{R\_m'} \\ \mathcal{L}_{VAE} &= \mathrm{KL}(E(m', r) \| p(t|r)) - \mathbb{E}_{E(m', r)}[\log G(t, r)], \end{aligned} \tag{36}$$

where $p(t|r)$ is considered to be $\mathbb{N}(0, 1)$, $\mathcal{L}_{KL}$ is the Kullback-Leibler divergence, and $\mathcal{L}_{R\_m'}$ is the loss computed during the reconstruction of visual features denoted by $-\log G(t, r)$. Conversely,

f-WGAN consists of a discriminator $D(m', r)$, referred to as D, and generator $G(t, r)$. From a random input noise $t$, the generator $G(t, r)$ generates a visual feature $\hat{m}$ constrained by the semantic embedding $r$. In contrast, the discriminator takes a synthesized visual feature $\hat{m}$ or a real visual feature $m'$, which is also constrained by semantic embedding $r$ and results in a real value between 0 and 1. Optimization of f-WGAN loss is as follows:

$$\mathcal{L}_{WGAN} = \mathbb{E}[D(m', r)] - \mathbb{E}[D(\hat{m}, r)] - \mu\mathbb{E}[(\|\nabla D(\tilde{m}, r)\|_2 - 1)^2], \tag{37}$$

where $\tilde{m} = \rho m' + (1 - \rho m')$ with $\rho \sim \mathrm{U}(0, 1)$, and $\mu$ is the penalty coefficient.

## 2.6 FEATURE REFINEMENT (FR) MODULE

The feature refinement module (Chen et al., 2021b) in SuperCAT aims to enhance the visual features of ZSRSSC benchmarks. The triplet center margin and semantic loop consistency losses condition the FR module.

*Triplet Center Margin loss (TCM-loss):* This loss is designed to achieve discriminative features by pushing features with the same class label close together and features with different class labels far apart. It aims to achieve within-class similarity and between-class separability. This is typically accomplished using class label information, center loss Wen et al. (2016), and triplet loss Schroff et al. (2015). The $\mathcal{L}_{TCM}$ can be formally outlined as follows:

$$\mathcal{L}_{TCM}(\hat{r}, e, e') = max\left(0, \Gamma + \psi\|\gamma - c_e\|_2^2 - (1 - \psi)\|\gamma - c_{e'}\|_2^2\right) \tag{38}$$

where $c_e$ is the $e^{th}$ class centre of semantic embedding, $c_{e'}$ is the $e'^{th}$ class centre, $\Gamma$ refers the margin to handle the distance between the pairs of inter and intra class, $\gamma$ specifies the intermediate features in FR, and $\psi \in [0, 1]$ is utilized to balance the within-class similarity and between-class separability.

*Semantic Loop Consistency loss (SLC-loss):* We aim to reconstruct the semantic features $\hat{r}$ from the visual feature $m'$ or synthesized visual feature $\hat{m}$ with the help of the reparameterization trick(Kingma & Welling, 2013). The $\mathcal{L}_{R\_r}$ is applied to the reconstructed semantic features in the FR to ensure that synthesized semantic features $\hat{r}$ are transformed into the exact embeddings that generated them. By utilizing the $l_1$ reconstruction loss, the SLC loss is attained, formulated as follows:

$$L_{R\_r} = \mathbb{E}[\|\hat{r}_{real} - r\|_1] + \mathbb{E}[\|\hat{r}_{syn} - r\|_1], \tag{39}$$

where $\hat{r}_{syn}$ denotes the semantically related features synthesized from $\hat{m}$ and $\hat{r}_{real}$ signifies the semantically related features synthesized from $m'$ using the FR. Notably, $\hat{r} = \hat{r}_{real} \cup \hat{r}_{syn}$ and $r$ denotes the semantic embeddings for the given visual features $\hat{m}$ or $m'$.

*Extracting Fully Enhanced Features:* After training the FR module, we extract fully enhanced features $\tilde{m}_u$ and $\tilde{m}_s$ from the FR. We concatenate the visual features $m'$, corresponding latent representation $l_s \in L$ and semantic embedding $\tilde{r}_s \in R$ as $\tilde{m}_s$ using the residual connection. Similarly, we concatenate the visual features $\hat{m}$, corresponding latent representation $l_u$ and the semantic embedding $\tilde{r}_u$ to obtain $\tilde{m}_u$. The final enhanced features $\tilde{m}_s$ and $\tilde{m}_u$ are expressed as

$$\tilde{m}_s = m' \odot l_s \odot \hat{r}_s, \tag{40}$$

$$\tilde{m}_u = \hat{m}_u \odot l_u \odot \hat{r}_u, \tag{41}$$

where $\odot$ denotes concatentation operation, $\tilde{m}_s$ and $\tilde{m}_u \in \tilde{M}$.

The refined visual features $\tilde{m}_s$ and $\tilde{m}_u$ are designed to be discriminative, helping to reduce ambiguities across samples from different classes. The overall objective function of SuperCAT is defined as:

$$\mathcal{L}_{total} = \mathcal{L}_{CAT} + \mathcal{L}_{VAE} + \mathcal{L}_{WGAN} + \lambda_{TCM}\mathcal{L}_{TCM} + \lambda_{R\_r}\mathcal{L}_{R\_r}, \tag{42}$$

where $\lambda_{TCM}$ and $\lambda_{R\_r}$ are hyperparameters that control their corresponding loss functions.

*Zero-Shot Scene Classification:* In the refined feature space, we train a supervised classifier as the final classifier. For conventional zero-shot learning, the objective is to learn the classifier $f_{czsl} : \tilde{M} \to N_u$. During testing, the unseen test features are refined into new features by the FR module and used for classification.

## 3 EXPERIMENTAL RESULTS

In this section, we provide the quantitative and qualitative analysis of our SuperCAT framework on three benchmark datasets for scene classification in remote sensing images.

### 3.1 DATASETS

We utilize the semantic attributes Rambabu et al. (2024) for three benchmark scene classification datasets in remote sensing, namely, UCMercedLandUse (UCM21) (Yang & Newsam, 2010), Aerial Image Dataset (AID30) (Xia et al., 2016), and NWPU-RESISC45 (NWPU45) (Cheng et al., 2017a). Table 1 provides the details of each dataset. We have evaluated our SuperCAT framework for the CZSL setting using top-1 classification accuracy (Xian et al., 2017c).

Table 1: Details of scene classification datasets.

| Parameters | UCM21 | RS19 | AID30 | NWPU-RESISC45 |
|---|---|---|---|---|
| Number of scene classes | 21 | 19 | 30 | 45 |
| Samples per each class | 100 | 50 | 220-420 | 700 |
| Number of samples | 2,100 | 950 | 10,000 | 31,500 |
| *Number of semantic attributes* | *33* | *26* | *44* | *57* |

### 3.2 IMPLEMENTATION DETAILS

For zero-shot remote sensing scene classification, we utilize features of size $2048$ extracted from the ResNet-101 model, pre-trained on ImageNet without fine-tuning. In the SuperCAT framework, we set the learning rate, weight decay, and momentum to 0.0001, 0.0001 & 0.9, respectively, in the SGD optimizer with a batch size of 64. We use Adam optimizer (Kingma & Ba, 2014) by setting $\beta_1 = 0.5$ and $\beta_2 = 0.999$ values. We set $\lambda_{AR}, \lambda_{SC}, \lambda_{VSAT}, \lambda_{SCL_f}, \lambda_{SCL_p}$ to $\{0.01, 1.0, 0.01, 0.0001, 0.001\}$ for all datasets based on empirical analysis. The value of the penalty multiplier ($\eta$) is 10. In the FR module, our experiments consider 0.5 and 0.999 values to $TCM$ loss multiplier and $SLC$ loss multiplier. The balancing factor psi ($\psi$) is set to 0.4 for all the datasets.

### 3.3 ANALYSIS OF CLASSIFICATION PERFORMANCE USING SUPERCAT

Tables 2, 3, 4 show that our SuperCAT consistently outperforms state-of-the-art approaches across standard seen/unseen class splits (Li et al., 2022) on the UCM21, AID30, and NWPU45 datasets, respectively.

Table 2: Top-1 classification accuracy and standard deviation (%) on UCM21 dataset.

| Methods | 16/5 | 13/8 | 10/11 | 7/14 |
|---|---|---|---|---|
| VSC (Wan et al., 2019) | 55.91 ± 11.77 | 36.26 ± 07.31 | 25.97 ± 05.79 | 19.53 ± 03.05 |
| f-CLSWGAN (Xian et al., 2017b) | 56.97 ± 11.06 | 36.47 ± 06.28 | 27.89 ± 04.99 | 19.34 ± 03.96 |
| DSAE (Wang et al., 2021) | 58.63 ± 11.23 | 37.50 ± 07.79 | 25.59 ± 05.24 | 20.18 ± 03.07 |
| CSPWGAN (Li et al., 2022) | 62.66 ± 10.79 | 46.19 ± 05.52 | 35.17 ± 04.93 | 26.17 ± 03.87 |
| RSZero-CSAT (Rambabu et al., 2024) | 71.40 ± 10.90 | 49.10 ± 06.20 | 38.30 ± 04.97 | 26.70 ± 03.60 |
| **SuperCAT (ours)** | **73.35 ± 10.45** | **52.40 ± 05.25** | **39.51 ± 04.47** | **29.13 ± 03.07** |

Table 3: Top-1 classification accuracy and standard deviation (%) on AID30 dataset.

| Methods | 25/5 | 20/10 | 15/15 | 10/20 |
|---|---|---|---|---|
| VSC (Wan et al., 2019) | 52.61 ± 08.37 | 35.85 ± 05.52 | 26.11 ± 03.76 | 17.50 ± 02.19 |
| f-CLSWGAN (Xian et al., 2017b) | 50.68 ± 11.25 | 33.89 ± 05.72 | 24.95 ± 02.96 | 17.26 ± 03.06 |
| DSAE (Wang et al., 2021) | 53.49 ± 08.58 | 35.32 ± 05.17 | 25.92 ± 03.92 | 17.65 ± 02.52 |
| CSPWGAN (Li et al., 2022) | 55.86 ± 10.60 | 37.93 ± 05.26 | 26.97 ± 02.53 | 19.43 ± 03.02 |
| RSZero-CSAT (Rambabu et al., 2024) | 66.90 ± 10.24 | 41.81 ± 05.36 | 31.30 ± 03.10 | 23.60 ± 02.89 |
| **SuperCAT (ours)** | **69.80 ± 09.72** | **45.22 ± 05.33** | **32.30 ± 02.39** | **24.09 ± 02.64** |

Table 4: Top-1 classification accuracy and standard deviation (%) on NWPU45 dataset.

| Methods | 35/10 | 30/15 | 25/20 | 20/25 |
|---|---|---|---|---|
| VSC (Wan et al., 2019) | 50.68 ± 06.60 | 40.92 ± 04.59 | 30.62 ± 03.10 | 25.51 ± 02.04 |
| f-CLSWGAN (Xian et al., 2017b) | 56.97 ± 11.06 | 36.47 ± 06.28 | 27.89 ± 04.99 | 19.34 ± 03.96 |
| DSAE (Wang et al., 2021) | 51.22 ± 06.91 | 41.94 ± 04.61 | 31.85 ± 03.32 | 25.20 ± 02.17 |
| CSPWGAN (Li et al., 2022) | 50.66 ± 05.86 | 41.61 ± 04.48 | 32.09 ± 02.96 | 26.65 ± 02.33 |
| RSZero-CSAT (Rambabu et al., 2024) | 56.80 ± 06.23 | 44.90 ± 04.67 | 36.60 ± 03.00 | 26.20 ± 02.43 |
| **SuperCAT (ours)** | **57.57 ± 05.75** | **46.18 ± 04.46** | **38.69 ± 02.24** | **28.45 ± 02.27** |

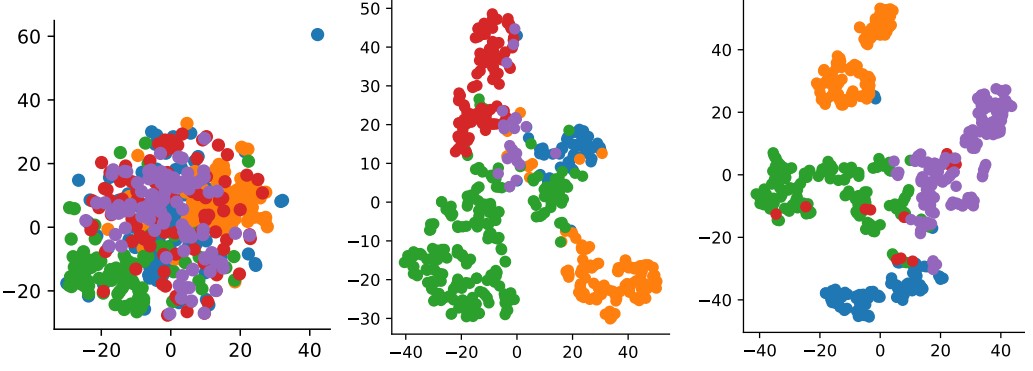

(a) Test unseen class CNN features (b) Test unseen class CAT features (c) Test unseen class SuperCAT features (Accuracy 63.3%) tures (Accuracy 73.4%)

Figure 2: The t-SNE visualizations of visual features for the unseen classes from the UCM21 dataset.

### 3.4 QUALITATIVE ANALYSIS OF SUPERCAT

Figure 2 illustrates the qualitative analysis of CAT and FR modules of SuperCAT on the UCM21 dataset. We employ t-distributed stochastic neighbour embedding (t-SNE) van der Maaten & Hinton (2008) to depict the visual features of the CNN backbone, visual features after the CAT module and the refined visual features obtained after the FR module in the SuperCAT framework over a randomly selected five unseen class samples. Figure 2b shows the visual features obtained from the CAT module. From Figure 2b, we can observe that the visual features obtained from the CAT module are separable under the supervision of softmax loss. However, these are not discriminative enough for label prediction of unseen classes in remote sensing scene classification, as they exhibit significant intra-class variations. Therefore, directly using these features for recognition may not be suitable. Figure 2c shows the clear separability of our proposed method and efficient label prediction of visual features with the FR module. The results indicate a cumulative contribution of super-resolution, CAT, f-VAEGAN, and FR modules in our SuperCAT in achieving a discriminative semantic space by capturing the meaningful semantics pertinent to unseen classes.

## 4 CONCLUSION

This paper proposes a SuperCAT framework for classifying the scenes in remote-sensing images by combining a super-resolution module with the zero-shot scene classification example. The Super-CAT leverages the semantic attributes to explore the semantic knowledge of unseen & seen classes. It comprises a cross-semantic attribute-guided Transformer (CAT) module, a feature-generating model (f-VAEGAN), a feature refinement (FR) module, and a classifier. The CAT module is proposed to extract visual features guided by semantic attributes and semantic attribute features guided by visual features. We use an f-VAEGAN in SuperCAT to generate synthetic features for unseen classes constrained by semantic vectors. Further, we employ an FR module to effectively refine the visual features and improve the precise classification of unseen & seen class remote sensing samples. Our extensive experiments on three benchmark remote sensing image scene classification datasets show the efficacy of SuperCAT over state-of-the-art approaches.

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
