# SuperCAT: Super Resolution and Cross Semantic Attribute-guided Transformer based Feature Refinement for Zero-Shot Remote Sensing Scene Classification

## 1 Contribution of Model Components and Loss Functions

To delve deeper into SuperCAT, we have conducted ablation studies to evaluate the *contribution* of the cross-semantic attribute-guided Transformer (CAT) module, super-resolution module, triplet-centre margin loss ($\mathcal{L}_{TCM}$) and the semantic loop consistency loss ($\mathcal{L}_{SLC}$). Our results are presented in Table 1. In particular, when the CAT module is excluded, the performance of SuperCAT significantly declines compared to its full model. Specifically, the accuracy drastically decreased by 10.05% on UCM21 and by 8.6% on AID30, respectively. Further, with the inclusion of $\mathcal{L}_{TCM}$ in SuperCAT, the mean classification accuracy of 2.35% and 2.7%, respectively, is substantially improved for UCM21 and AID30 datasets. The performance is further improved by integrating the $\mathcal{L}_{SLC}$ into our model. With the inclusion of super-resolution in SuperCAT, the mean classification accuracy of 1.25% and 1.9% is substantially improved for UCM21 and AID30 datasets, respectively.

Table 1: Top-1 classification accuracy on the SuperCAT model without including CAT and super-resolution modules and loss functions.

| $Method$ | UCM21 | AID30 |
|---|---|---|
| CAT | 63.3 | 61.2 |
| SuperCAT w/o $\mathcal{L}_{SLC}$ | 71.0 | 67.1 |
| SuperCAT w/o $\mathcal{L}_{TCM}$ | 71.4 | 67.4 |
| SuperCAT w/o super-resolution | 72.1 | 67.9 |
| SuperCAT (full) | 73.35 | 69.80 |

## 2 Analysis of Semantic Attributes on SuperCAT

We have conducted ablation studies on SuperCAT with and without semantic attributes. We employ word embeddings to evaluate SuperCAT without considering semantic attributes. Table 2 provides the quantitative analysis of SuperCAT regarding classification accuracy for both seen and unseen class samples of remote sensing images. It is observed from Table 2 that our SuperCAT can classify the unseen categories better than other scenarios. Here, $S$ indicates seen class accuracy, $U$ indicates unseen class accuracy, and $H_m$ indicates harmonic mean class accuracy calculated as $2(S \times U)/(S + U)$.

Table 2: Analysis of the SuperCAT with semantic and without semantic attributes.

| $Method$ | UCM21 | | | |
|---|---|---|---|---|
| | $ACC$ | $S$ | $U$ | $H_m$ |
| FREE Chen et al. (2021)-word2vec (w/o fine-tuning) [SuperCAT w/o CAT] | - | 66.6 | 30.8 | 44.5 |
| FREE Chen et al. (2021)-attr values (w/o finetuning) [SuperCAT w/o CAT] | - | 51.6 | 38.2 | 45.2 |
| SuperCAT (ours) | 73.4 | 74.3 | 56.3 | 64.1 |

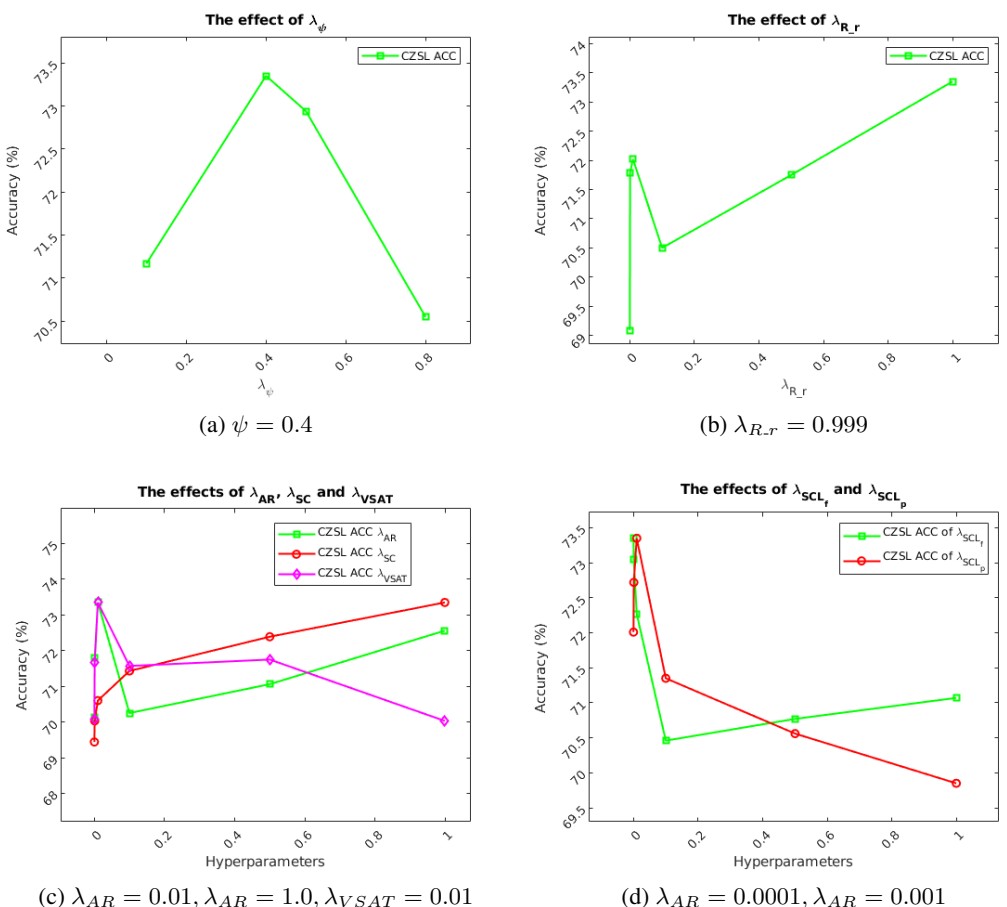

Figure 1: The effect of hyperparameters on UCM21 dataset.

# 3 SIMILARITY OF CAT AND FR MODULE REPRESENTATIONS

We calculated the similarity between the representations of the CAT and FR modules of SuperCAT on the UCM21 and AID30 datasets using centered kernel alignment (CKA) Kornblith et al. (2019) in the CZSL setting. The results in Table 3 indicate that the similarity between the feature representations from the trained CAT and FR modules is significantly less, as the SuperCAT framework refines the visual features to classify unseen categories better.

Table 3: Similarity of CAT and FR representations on the UCM21 and AID30 datasets.

| Visual Features | Similarity Index UCM21 | Similarity Index AID30 |
|---|---|---|
| Test unseen class features from CAT and FR modules in CZSL setting | 0.58564 | 0.65432 |

# 4 HYPERPARAMETER ANALYSIS

We study the impact of the balance factor $\psi$ on the FR module. As Figure 1a illustrates, the growth of $\psi$, the $ACC$ consistently improves on the UCM21. This demonstrates an enhancement in intra-class closeness and inter-class distinctiveness. Larger gains in intra-class closeness are observed when classes are confused, while improved inter-class distinctiveness significantly benefits the classification of ambiguous classes. We set $\psi$ to 0.4 for the CZSL setting for all the datasets.

We analyzed the FR module's hyperparameter $\lambda_{R\_r}$ of semantic loop consistency loss. It is observed that the improvement in the classification accuracy is achieved at $\lambda_{R\_r} = 0.999$ for the CZSL

setting for all the datasets. We have provided the analysis for the UCM21 datasets in figure 1b. We have also analyzed hyperparameters $\{\lambda_{AR}, \lambda_{SC}, \lambda_{VSAT}, \lambda_{SCL\_f}, \text{and} \lambda_{SCL\_p}\}$ (1c, 1d), set to $\{0.01, 1.0, 0.01, 0.0001, 0.001\}$, respectively.

## 5 ANALYSIS OF MODEL EFFICIENCY

We have calculated efficiency in terms of computational cost (GPU occupied (GB) and time per step(s) during model training), which are provided in the Tables 4, 5, and 6.

Table 4: Analysis of computational efficiency of super-resolution module

| Dataset | GPU occupied | Time per step(s) |
|---|---|---|
| UCM21 | 26.394 GB/batch of 50 samples | 3.80037 seconds/sample |
| AID30 | 25.998 GB/batch of 45 samples | 71.21889 seconds/sample |
| NWPU45 | 26.396 GB/batch of 50 samples | 3.80466 seconds/sample |

Table 5: Analysis of computational efficiency of CAT module

| Computation Cost | Without Super-Resolution | With Super-Resolution |
|---|---|---|
| GPU occupied | 4.734 GB / batch of 50 training samples | 4.734 GB / batch of 50 training samples |
| Time per step(s) | 0.177672 seconds/iteration | 0.205013 seconds/iteration |

Table 6: Analysis of computational efficiency of FR module

| Computation Cost | Without Super-Resolution | With Super-Resolution |
|---|---|---|
| GPU occupied | 4.742 GB /batch of 50 training samples | 4.742 GB /batch of 50 training samples |
| Time per step(s) | 12.9007 seconds/batch of 50 training samples | 17.5325 seconds/batch of 50 training samples |

## 6 SUPER RESOLUTION IMAGES

We employ an efficient diffusion model, Resshift Yue et al. (2023), for super-resolution to obtain high-resolution images from low-resolution. Figure 2 depicts some super-resolution images of remote sensing samples obtained from the ResShift model.

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

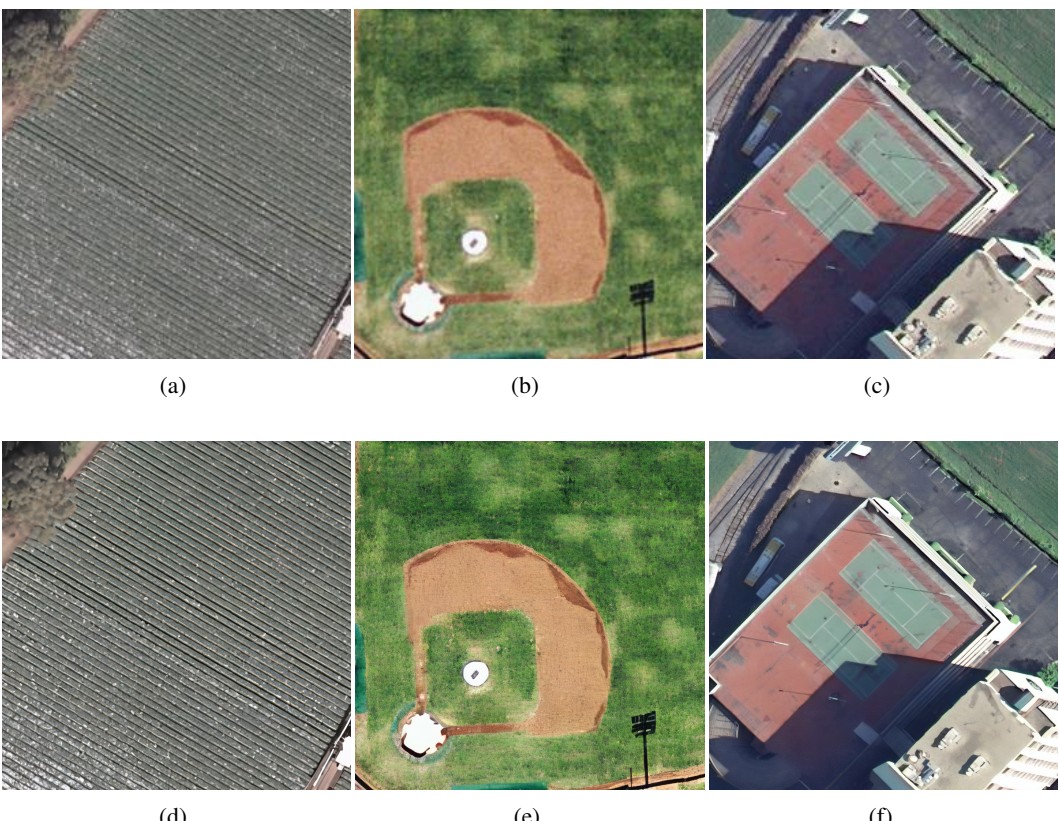

|  |  |  |
|:---:|:---:|:---:|
| (a) | (b) | (c) |
| (d) | (e) | (f) |

Figure 2: Some of the super-resolution images of remote-sensing samples. The first row depicts the samples of remote sensing images. The second row shows the super-resolution images of remote sensing samples obtained from the ResShift model.