# OpenReview forum: "SuperCAT: Super Resolution and Cross Semantic Attribute-guided Transformer based Feature Refinement for Zero-Shot Remote Sensing Scene Classification"
_ICLR.cc/2025/Conference — ICLR 2025 Conference Withdrawn Submission_

### Official Review · Reviewer_7KS6 · 2024-11-01

**Soundness:** 3
**Presentation:** 3
**Contribution:** 2
**Rating:** 3
**Confidence:** 5

**Summary:**

This paper introduces SuperCAT, a novel framework for zero-shot remote-sensing scene classification (ZSRSSC). SuperCAT innovatively combines super-resolution with classification to enhance performance on remote sensing images. It comprises a super-resolution module, a cross-semantic attribute-guided Transformer (CAT), feature-generating models, and a feature refinement (FR) module. These components work together to overcome the challenges of extracting discriminative semantic knowledge from scenes with complex variations and non-uniform spatial resolutions, demonstrating promising experimental results.

**Strengths:**

Overall, the structure of this paper is clear, and the motivation is well-defined. It performs excellently in setting the background and framing the problem, providing a good overview of the literature on the topic. The technical aspects are also well explained, making it easy to understand how the algorithm works and is implemented.

**Weaknesses:**

A substantive assessment of the weaknesses of the paper. Focus on constructive and actionable insights on how the work could improve towards its stated goals. Be specific, avoid generic remarks. For example, if you believe the contribution lacks novelty, provide references and an explanation as evidence; if you believe experiments are insufficient, explain why and exactly what is missing, etc.

The novelty of the method is insufficient, as the proposed modules are largely based on existing approaches. The core module, CAT, bears a close resemblance to RSZero-CSAT.  The descriptions of the CAT module in sections 2.2 to 2.4 of this paper are largely consistent with those of RSZero-CSAT, including both the network structure and formula processing. Any unique designs should be thoroughly detailed, along with an explanation of their significance, while minimizing overlap with descriptions from other papers.

More concerns can be found in Questions.

**Questions:**

1.	Which modules are your innovative designs, and what are their respective functions?
2.	Some content lacks explanation, such as the definition of "semantic attributes" and the significance of "16/5" in Table 2. It would be beneficial to define "semantic attributes" upon their first introduction and to clarify the meaning of the split ratios in the table captions or within the accompanying text discussing Table 2.
3.	Figure 1 depicts the detailed structures of SAVT and VSAT. Although the diagram shows that their structures are fundamentally similar, the two serve entirely different functions. The distinction that is not conveyed in the figure. For SAVT, the encoder layer outputs$$$I_{aug}^{a\to v}(m)$ are used as inputs to multi-head cross attention, as keys ($K_{t}^{a\to v}$), values ($V_{t}^{a\to v}$), and queries ($Q_{t}^{a\to v}$) to be obtained as semantic embeddings ${{\mathcal{R}}_{A}}$. In contrast, for VSAT, $Q_{t}^{v\to a}$is obtained from $I_{aug}^{v\to a}(m)$. Clearly distinguishing the Q, K and V of SAVT and VSAT in the figure would enhance its comprehensibility.
4.	Table 2 in the supplementary materials presents ablation experiments related to semantic attributes. As noted in Section 2: “We obtain attribute vectors for each attribute${{\mathcal{R}}_{A}}=\{{{r}_{a}}\}_{a=1}^{A}$ using the word2vec model”, the attribute values are derived from word2vec. Therefore, the first two entries in the table should be identical. How should this discrepancy be interpreted?

---

### Official Review · Reviewer_wmmV · 2024-11-03

**Soundness:** 2
**Presentation:** 2
**Contribution:** 2
**Rating:** 5
**Confidence:** 4

**Summary:**

This paper focuses on the challenge of identifying unknown samples in remote sensing image scene recognition and attempts to address this issue through zero-shot learning. Specifically, the authors propose a SuperCAT framework, which consists of four modules: a super-resolution module, a cross-semantic attribute-guided Transformer (CAT), feature-generating models, and a feature refinement (FR) module. Experimental results validate the effectiveness of the proposed method.

**Strengths:**

This paper addresses a highly challenging problem and achieves promising experimental results through a modular design approach.

**Weaknesses:**

The authors released a link to their code that is not anonymous, revealing the author’s identity, which violates ICLR's double-blind review policy.

The four components included in the proposed method are established and widely used techniques. However, the authors have not explained the rationale for combining these components or how each component, either individually or in combination, can enhance the performance of remote sensing image recognition.

The motivation for this paper's proposed method is unclear. For example, why are semantic attributes used to model the distinct characteristics of diverse scenes in zero-shot scene classification?

The authors dedicate a substantial amount of space (6 pages) to introducing the proposed method. It is recommended to move the descriptions of existing methods to the supplementary material.

The experimental analysis is inadequate, lacking component contribution analysis, discussion of limitations, analysis of semantic attributes, and hyperparameter analysis.

The paper lacks a comparison with state-of-the-art (SOTA) methods and does not sufficiently survey recent work from the past two years.

**Questions:**

Please refer to my comments.

---

### Official Review · Reviewer_yU9P · 2024-11-09

**Soundness:** 2
**Presentation:** 2
**Contribution:** 2
**Rating:** 3
**Confidence:** 3

**Summary:**

The paper proposes a framework called SuperCAT for zero-shot scene classification. Specifically, it focuses on classifying the scenes in remote-sensing images. The framework consists of several modules such as:
1) Super Resolution Module is used to address low-resolution challenges for remote-sensing images,
2) Cross-Semantic Attribute-Guided Transformer (CAT) has two modules called SAVT and VSAT modules aligns semantic and visual spaces for feature learning,
3) Feature Generation with f-VAEGAN synthesizes visual features constrained by semantic vectors to enhance zero-shot capabilities, and
4) Feature Refinement (FR) module improves class separability using Triplet Center Margin and Semantic Loop Consistency Losses.

**Strengths:**

1. The proposed method leverages a diffusion model (ResShift) to address low-resolution issues in remote sensing data, improving image quality and facilitating richer feature extraction.
2. The proposed Feature Refinement (FR) module enhances class separability by improving Intra-Class compactness and Inter-Class distinctiveness.

**Weaknesses:**

The majority of the work presented in this paper has already been published in the "2024 International Joint Conference on Neural Networks (**IJCNN**)" titled **RSZero-CSAT: Zero-Shot Scene Classification in Remote Sensing Imagery using a Cross Semantic Attribute-guided Transformer.** Only minor modifications, such as incorporating a diffusion model, a softmax layer, and FR module were made.

1. The proposed framework (Figure 1), utilized in the RSZero-CSAT paper, with the exception of the super-resolution module and the softmax layer, which is a new component and not cited appropriately, is depicted in Figure 1.
2. Almost most of the information from following sections 2.2.1, 2.3, 2.4 (or most of the sections) draws heavily from the RSZero-CSAT paper, including equations, and sentences that are not cited properly.

**Visualization Issues:**
1. The figure lacks a clear legend labeling. In figure 1(a) elements like "**CNN Backbone**" are mentioned but are not clearly referenced or located within the image.
2. There are also unexplained characters, such as the symbol “**j**” near "Visual → Semantic Attribute Transformer" and "Semantic Attribute → Visual Transformer"

**Performance Issues:**

1. In **Figure 2(c)**, the visualization of class separability in the refined feature space includes only four class samples, with the red class samples merged into remaining four class samples, unlike **Figures 2(a) and 2(b)**, which both show five classes. This inconsistency raises concerns about the model’s ability to maintain class separability.

2. Even though model is achieving good performance over the existence methods, the model’s complexity requiring multiple loss functions, transformers, and super resolution may make this framework computationally expensive.This may limit practicality. For example, the GPU memory usage and time-per-sample are relatively high, especially for the super-resolution module for AID30.

3. In Table 1 of supplementary material, it is shown that after adding an efficient super-resolution module, there is a 1.25% and 1.9% improvement on the UCM21, and AID30 dataset but with high-computational demands indicated in Table 4 for AID30 which consumes **71.22 seconds per sample**. This discrepancy raises questions about the practical value of the proposed super-resolution module.

**Questions:**

1. Given the high memory usage and processing time reported in the supplementary material (e.g., up to **71.22 seconds per sample** for AID30 in super-resolution), have you considered any alternatives to reduce computational costs?
2. In **Figure 2(c)**, there are only four class samples, with the red class samples merged into other class samples. Does this merged class indicate a limitation in the FR module’s class separability?
3. In **Figure 3(d) and 3(e)**, the class separability in **RSZero-CSAT** appears more separated for unseen samples without the inclusion of the diffusion model, softmax layer, and FR module. Could you explain why RSZero-CSAT achieves this level of separation, and what specific advantage SuperCAT offers in terms of class separability given its additional components?

**Details Of Ethics Concerns:**

A lot of the content in sections 2.2.1, 2.3, and 2.4 (and other sections) seems to be heavily drawn from the RSZero paper, with entire sentences directly copy-pasted from the published text.

---

### Note · Authors · 2024-11-14

I have read and agree with the venue's withdrawal policy on behalf of myself and my co-authors.